# Autonomous 6-DOF Manipulator Operation for Moving Target by a Capture and Placement Control System

**DOI:** 10.3390/s22134836

**Published:** 2022-06-26

**Authors:** Xiang Chen, Peilin Liu, Rendong Ying, Fei Wen

**Affiliations:** School of Electronic Information and Electrical Engineering, Shanghai Jiao Tong University, Shanghai 200240, China; chenxiang_hit@sjtu.edu.cn (X.C.); rdying@sjtu.edu.cn (R.Y.); wenfei@sjtu.edu.cn (F.W.)

**Keywords:** autonomous operation, 6-DOF manipulator, capture and placement, velocity feedforward, refined PID

## Abstract

The robot control technology combined with a machine vision system provides a feasible method for the autonomous operation of moving target. However, designing an effective visual servo control system is a great challenge. For the autonomous operation of the objects moving on the pipeline, this article is dedicated to developing a capture and placement control system for the six degrees of freedom (6-DOF) manipulator equipped with an eye-in-hand camera. Firstly, a path planning strategy of online capture and offline placement is proposed for real-time capture and efficient placement. Subsequently, to achieve the fast, stable, and robust capture for a moving target, a position-based visual servo (PBVS) controller is developed by combining estimated velocity feedforward and refined PID control. Feedforward control is designed using the estimated velocity by a proposed motion estimation method for high response speed. PID control is refined by dead zone constraint to reduce the manipulator’s jitter caused by the frequent adjustment of manipulator control system. Besides, the proportional, integral, and differential coefficients of PID controller are adaptively tuned by fuzzy control to reject the noise, disturbance, and dynamic variation in the capture process. Finally, validation experiments are performed on the constructed ROS–Gazebo simulation platform, demonstrating the effectiveness of the developed control system.

## 1. Introduction

With the development of artificial intelligence, the manipulator control technology combined with visual feedback has developed rapidly and attracted much attention in the fields of assembling parts, sorting materials, and capturing objects, etc. Especially, the autonomous operation for the objects moving on the pipeline in industrial production or warehousing logistics is a great challenge. The inaccurate or unstable operation will lead to high costs and even damage to manipulator [1,2,3]. On one hand, the precise knowledge of the moving target needs to be obtained by machine vision technology. On the other hand, an effective control system should be developed based on the feedback of visual system to adjust the manipulator’s pose and motion. It is impossible to cover all the above deeply in a single article, so the issues related to control are mainly studied here.

Generally, visual servo controller can be classified as position-based, image-based, and hybrid system according to the errors used in a controller [4,5]. Position-based visual servo (PBVS) controller adjusts the error between the desired and actual pose and motion in 3D workspace directly. Image-based visual servo (IBVS) controls a manipulator by the deviation between the measured and target’s position on 2D image plane. However, additional measurement is required because IBVS lacks the depth information. Although hybrid visual servo is effective, it is more complex than PBVS and IBVS. This will reduce the real-time performance of a manipulator system. Compared with IBVS and hybrid system, PBVS has a simpler control framework that allows direct control for end-effector’s pose and motion, while the disadvantages are that moving target’s information are susceptible to camera calibration and image noise. At present, these difficulties have been successfully solved by researchers, such as the image errors caused by the uncalibrated camera [6,7] and the image noise generated by camera vibration during manipulator operation [8].

Owing to the advantages of follow-up view and the natural relationship of the end-effector’s pose relative to target, PBVS equipped with an eye-in-hand camera has attracted much attention from the researchers. A visual servo control scheme based on Kalman filter was proposed to automatically capture the moving target, which presented good robustness under noise and unexpected disturbance [9]. The dual Kalman filter scheme was developed to improve the tracking robustness and provide smooth motion estimation for the PBVS robotic control system [10]. A pose and motion estimation algorithm was proposed based on photogrammetry and extended Kalman filter, and the PBVS controller was devised based on PID algorithm to capture the non-cooperative target [11]. The inverse kinematics method incorporated trajectory planning was proposed based on the virtual repulsive torque theory. It was applied to design a PBVS system, and the manipulator’s trajectory approaching target was spontaneously determined by the output control law [12]. However, due to the nonlinearity, time-varying parameters and model uncertainty of manipulator, and the high requirements for efficiency, accuracy, and robustness in practical engineering, the current control methods can hardly meet the control needs.

Some studies have been conducted to improve the performance of manipulator control system. Sliding mode controller was developed using the nonlinear model of manipulator for high robustness [13,14]. However, this controller was implemented by modeling the manipulator system and was hard to achieve precise control due to the nonlinear characteristics of manipulator. To address the drawback, the neural network-based controller is being studied, such as the back-propagation neural network used for reducing manipulator’s tracking error [15], recurrent neural network controller [16,17], reinforcement learning neural network controller for the manipulator with unknown parameters and dead zones [18], hybrid neural network and sliding mode control for manipulator in dynamical environment [19]. Although neural network-based control theory obtains good results, how to find the optimal solution and achieve the fast convergence rate need to be solved in control engineering. In contrast, the controllers combining PID and other optimization or control methods are widely used in nearly 90% of industrial control systems due to their advantages of simple framework, good stability, and high reliability [20,21,22]. The whale optimizer algorithm is applied to adjust the parameters of PID controller, obtaining less settling time and ITAE error during the trajectory tracking control of 2-DOF robot manipulator [23]. A fuzzy PID controller was used to design the kinematics control strategy of cable-driven snake-like manipulator, realizing the precise control [24]. The following error will increase when inputting rapidly changed signals for the PID-based controller. To overcome this drawback, the feedforward plus PID controller was developed and achieved better tracking results and less tuning time in the comparative experiments [25]. The control system combined feed-forward, and fuzzy PID controller was proposed to control a planar parallel manipulator, which showed good adaptability to the disturbances and dynamic variations [26].

Based on the above, it can be concluded that two problems need to be addressed in order to design an effective autonomous operation control system for the moving target on pipeline. One is how to raise the overall operation efficiency, and the other is how to improve the response speed, stability, and robustness for capturing the moving target. In this paper, a capture and placement control system is developed, which is as follows: a path planning strategy of online capture and offline displacement is proposed to improve the overall operation efficiency; a new PBVS controller is developed to improve the capture performance by combining the estimated velocity feedforward and refined PID controller. The motion estimation method combined Kalman filter and interpolation operation is proposed to achieve accurate and smooth estimation for target’s pose and motion. Therefore, it provides feedforward velocity to the controller for a high response speed. The refined PID controller is designed, by the dead zone constraint, to reduce the manipulator’s jitter caused by controller’s frequent adjustment, and by fuzzy control to adaptively tune PID coefficients for rejecting noise or external disturbance. Finally, the developed capture and placement control system is experimentally verified by the constructed ROS–Gazebo simulation platform.

## 2. ROS–Gazebo Simulation Platform

To facilitate the development of capture and placement control system, a simulation platform was built by combining robot operating system (ROS) and Gazebo simulator. All the experiments were performed on this platform. ROS is an open-source meta-operating system that provides many services for robots, including hardware abstraction, underlying device control, implementation of common functions, inter-process messaging, and package management [27]. It can be used to drive real or simulated robots to perform various tasks. Gazebo is a mature 3D physical simulator that can be closely combined with ROS for simulation [28]. To obtain high simulation accuracy, Gazebo provides various underlying physics engine such as Ode, Bullet, Dart, and Simbody. Besides, it has rich interfaces for modeling, control, and visualization. In the virtual visual simulation environment, the robot model with physical properties can be created, and the robot motion and sensing data can also be simulated. Compared with the direct hardware operation, the simulation platform based on ROS and Gazebo is more suitable for the development and test of complex system due to the convenient parameter adjustment, fast operation speed, and low cost. The simulation framework of ROS–Gazebo is shown in Figure 1a, including the following processes:(1)Gazebo updates the model state information in real time by the simulation model plugin;(2)ROS control layer regularly obtains and calculates the state information from the simulation model;(3)The behavior algorithm obtains the calculated state information and performs behavior derivation calculation;(4)The node of generating control instruction obtains the behavior information and generates the control instructions;(5)Gazebo simulation plugin gets control instructions to control the motion of simulation model.

The robot simulation model was built as presented in Figure 1b. It mainly consists of a 6-DOF manipulator equipped with an eye-in-hand camera and vacuum gripper, the object moving on pipeline and a material box. Thereinto, UR5 with 0.85 m operating radius and +/−0.1 mm repeatable accuracy was selected as the manipulator model. Kinect V1 with 640 × 4180 resolution and 30 fps are selected as the camera model. The accurate and efficient simulation for the capture and placement of the moving target can be achieved by Gazebo simulator due to its powerful physics engine. Thereinto, the vacuum gripper will perform capture when the tracking error of manipulator end-effector relative to target is within a certain range. The visual servo control program is developed by C++ and Python in ROS control layer.

## 3. Kinematic Model of 6-DOF Manipulator Capturing Moving Target

The position-based visual seroving manipulator with an eye-in-hand camera provides an effective method for the autonomous operation of moving target, due to its advantages of follow-up view and natural relationship of end-effector’s pose with respect to target. In the operation process, the velocity of moving target may be unknown or changed dynamically. It is required to track the target in real time based on the feedback of the machine vision system. Besides, soft grasping is preferred for avoiding the damage to moving target. Therefore, not only the pose but also the velocity of manipulator end-effector should be consistent with those of moving targets. It is necessary to analyze the kinematic relationship between the manipulator and moving target. In this paper, the autonomous operation for the target moving on pipeline is conducted by a 6-DOF manipulator considering its high flexibility, as shown in Figure 2.

The camera was mounted next to the end-effector of manipulator and used to measure target’s pose xwc relative to camera frame in real time. The calibrations, including the intrinsic parameters for camera, the extrinsic parameters for hand–eye, i.e., the transformation matrix Tce from camera to end-effector frame (denoted by dashed line in Figure 2) and the model parameters for manipulator, i.e., the transformation matrix Teg from end-effector to global frame (marked with dotted line in Figure 2), were carried out in advance. The manipulator has six revolute joints. The first three, including shoulder pan joint, shoulder lift joint, and elbow joint, were used to control the position of manipulator end-effector. The last three include wrist-1 joint, wrist-2 joint, and wrist-3 joint for controlling the orientation of manipulator end-effector. During the autonomous operation, manipulator end-effector starts from the fixed initial point *S* and tracks the target moving from the point *O* on pipeline. When the tracking error of manipulator end-effector from moving target is within the allowed range, the capture operation will be performed at a certain grasping point *G*. It should be noted that point *G* is not a fixed point, but the one that meets the requirement of the allowed tracking error. A double dashed line in Figure 2 presents the motion trajectory of the moving target.

The kinematics of manipulator presents the transformation relationship from joint space to workspace as expressed below:(1)xeg=Teg(θ)
where xeg denotes the end-effector’s pose in a global frame, i.e., the base coordinate system of manipulator, and $\theta =(\theta_{1},\theta_{2},\theta_{3},\theta_{4},\theta_{5},\theta_{6})$ represents the angle of six joints in joint space. The forward kinematics model of the manipulator is given in Appendix A. Target’s pose xwg in global frame can be obtained according to the pose xwc in camera frame by:(2)xwg=Teg(θ)Tcexwc

The purpose of PBVS controller is to minimize the error between the desired and the actual pose of manipulator end-effector, which is expressed as:(3)e(k)=xwg(k)−xeg(k)
where ***e***(*k*) denotes the error in the *k*-th control cycle, including the position and orientation error; xwg(*k*) and xeg(*k*) are the estimated pose obtained by the proposed motion estimation method, which will be presented in Section 4.2.1, and the actual pose of end-effector in the *k*-th control cycle, respectively.

While minimizing the error, manipulator end-effector moves at the same velocity as the moving target, which facilitates the soft capture. In this case, the velocity control mode of manipulator is preferred for the real time tracking of the moving target, thus the output of the developed controller should be defined as the end-effector’s velocity in workspace.

During the actual control of robotic manipulator, end-effector’s motion speed in workspace should be transformed to joint space. Assume that in the *k*-th control cycle, veg(k) is the end-effector’s velocity in global frame and it is output by the designed controller, including linear velocity vleg(k) and angular velocity vaeg(k) expressed as:(4)veg(k)=[vleg(k),vaeg(k)]T

Both the linear and angular velocity are limited to the maximum velocity range. Finally, instantaneous inverse kinematic are performed, as given in Appendix B to obtain the desired joint velocity as:(5)ω(k)=J−1veg(k)
where $\omega (k)=\{\omega_{1}(k),\omega_{2}(k),\omega_{3}(k),\omega_{4}(k),\omega_{5}(k),\omega_{6}(k)\}$ is the joint angular velocity and $J^{-1}$ is the pseudo inverse of Jacobian matrix. Based on the obtained joint angular velocity, the angle of each joint can be expressed as:(6)$$\theta_{i}(k)=\theta_{i}(k-1)+\omega_{i}(k)T,\;i=0,1,2,\ldots \ldots ,6$$
where $\theta_{i}(k)$ is the control angle of the *i*-th joint in the *k*-th control cycle and *T* is the control period. The angles and angular velocities of all joints are finally input to the underlying controller of the manipulator to perform tracking and capture.

## 4. Capture and Placement Control System

A capture and placement control system was developed for the 6-DOF manipulator to autonomously operate the moving target on pipeline. This control system consists of two parts: an operation strategy of online capture and offline placement is proposed to improve the overall operation efficiency; a new PBVS controller is developed based on estimated velocity feedforward and refined PID control. Details are presented below.

### 4.1. Capture and Placement Strategy

During the capture process of the target moving on pipeline in industrial production or warehouse logistics, the target’s velocity is usually unknown or dynamically changed. Therefore, the robotic manipulator is required to track the target in real time. At the placement stage, the placed pose of target is usually fixed, and thus the motion path of manipulator end-effector can be planned offline using the predefined waypoints. According to the requirement for system operation, a capture and placement strategy is proposed as shown in Figure 3. Manipulator end-effector starts tracking the moving target from the initial pose when it appears in the camera’s field of view. Subsequently, the grasping operation is performed when the tracking error between end-effector and moving target is within the allowed error threshold. The motion path of the above capture process is online, planned based on the visual feedback, and the velocity control mode of manipulator is adopted for high real-time performance. The velocity control signal of manipulator capturing the moving target is output by the developed controller, which is mainly studied in this paper.

The capture process will cost large amounts of computing resource by the machine vision and manipulator control system. However, the motion path of placement process can be offline planned based on the predefined waypoints, including pre-placing pose, placing pose, and post-placing pose, for reducing the computing cost. The schematic diagram of offline placement is shown in Figure 4. The target’s pose xwg in global frame is obtained by the proposed motion estimation method, and it will be given in Section 4.2.1. The raising pose $x_{rai}$ can be determined only by changing the *z* axis coordinate value of target’s pose. The placing pose of all targets in material box is predetermined according to the size of material box and target. In Figure 4, $w_{i}$ is the placing pose of the *i*-th target. At the same time, the pre-placing pose $w_{pre}$ can be determined only by modifying the *z* axis coordinate value of placing pose. The joint angle control signal for placement operation can be obtained by inverse kinematics based on the pose of the key waypoints. As a result, the placement process avoids the online calculation, which helps to improve the operation efficiency.

### 4.2. Capture Control Scheme

Although control theory has developed rapidly, PID control strategy is still widely used in the industrial control systems due to its advantages of simple control structure, good stability, and high reliability. To track the moving target in real-time, it is required to have high dynamic response speed for the controller. Using the large proportional coefficient can improve the dynamic response of PID controller, but it easily causes excessive torque in some joints, leading the manipulator to stop working or even suffer damage. Besides, the target’s velocity may be changed by external disturbance during the tracking process. To achieve the fast, stable, and robust capture for the moving target, a new PBVS control scheme is developed, as shown in Figure 5. For high response speed, the velocity feedforward control is designed based on the target’s velocity estimated by the proposed motion estimation method. For PID closed-loop control, the target’s pose estimated by the motion estimation method is input as the desired value and the actual one of manipulator end-effector is used as the feedback value. What is more, PID controller is improved by dead zone module to reduce manipulator’s jitter caused by the frequent adjustment of controller, and by fuzzy control module to adjust PID parameters online for rejecting the noise and disturbances. The feedforward velocity is combined with the output of the improved PID controller, which is used as the end-effector’s velocity ***V***. Finally, the joint velocity ω and angle θ can be obtained to conduct the capture operation. Details are as follows.

#### 4.2.1. Feedforward Control by the Motion Estimation Method

The velocity of moving target on pipeline may be unknown or variable in the dynamic environment. Besides, the control instructions sent to manipulator are generally required at 50 Hz frequency. However, the real-time performance of issuing control commands is directly affected by machine vision system. According to the literature [29], most of the current machine vision system is time-consuming, even using the camera with high sampling frequency. As a result, the update frequency of control instructions is lower than the required one, which will cause the manipulator to move unevenly. To address the above problems, a motion estimation method combined Kalman filter and interpolation operation is proposed as shown in Figure 6. The target’s pose ***Z***(*k*) measured by machine vision system is input to perform motion estimation calculation, where *k* represents the current *k*-th cycle, and the other symbols in this section have the same meanings as those in Section 3. According to the transformation from camera to global frame and the back projection based on pinhole imaging principle, the measured pose ***Z***(*k*) of target in the global frame can be obtained by:(7)Z(k)=TegTceK−1U(k)s(k)
where *s*(*k*) is the distance from the optical center of camera to target. U(k) is the homogeneous coordinate value of target in pixel frame. ***K***^−1^ is the inverse matrix of camera intrinsic parameter matrix. Tce and Teg are the transformation matrix from camera to end-effector frame and from end-effector to global frame, respectively.

For the condition that ***Z***(*k*) is different from the previous ***Z***(*k* − 1), the Kalman filter is used to predict the upcoming measurement result. It computes the current estimation based on all past measurements and usually converges in a few iterations. Thus, the motion including velocity vwg(*k*), acceleration awg(*k*), and pose xwg(*k*) can be estimated by the same process, as described in literature [9]. One example of calculating the system state along *x* axis in global frame by Kalman filter method is given in Appendix C.

However, ***Z***(*k*) is usually not updated in time because the actual output frequency of machine vision system is lower than the required signal release frequency. As a result, the current sampled pose ***Z***(*k*) is equal with the previous ***Z***(*k* − 1). In this case, the interpolation operation will be conducted using the predicted velocity, acceleration, and position by (*k* − 1)-th Kalman filtering to estimate *k*-th motion, as expressed in Equation (8). From *k*-th to (*k* + *i* − 1)-th cycle, the interpolation operation is recursively performed, until receiving the updated measurement ***Z***(*k* + *i*). At this time, the interpolated results in (*k* + *i* − 1)-th cycle are applied to Kalman filtering process. The obtained velocity and position will be used as the feedforward value and the input for PID controller, respectively.
(8){vwg(k)=vwg(k−1)awg(k)=awg(k−1)xwg(k)=xwg(k−1)+vwg(k)*T+0.5*awg(k)*T2

Under the condition that the camera keeps stationary and the target moves on the pipeline plane, the comparative experiments are performed, in which all experimental data are represented in global coordinate system. Figure 7 gives target’s position measured by machine vision system based on the photogrammetry method and the ones by original Kalman filter and proposed motion estimation method. It is noted in Figure 7 that the measured data are composed of horizontal lines and polylines, which have the ladder-line shape. This discontinuity of measured data is caused by the update lag of target location, which will lead to the unstable movement of the manipulator. For the result obtained by the proposed method, phase a denotes the convergence process from the initial value to the exact one, and phase b indicates that the interpolated position achieves the stable and accurate estimation. The result estimated by original Kalman filter has the same adjustment process as the above, but it has poor smoothness and a larger difference with the measured result. According to the results, it can be concluded that the proposed method provides more smooth and accurate estimation for velocity feedforward and PID control.

#### 4.2.2. PID Controller with Dead Zone Constraint

In this paper, PID controller is used for adjusting the error ***e***(*k*) between the desired and actual pose of manipulator end-effector in steady-state tracking process. The estimated pose xwg(*k*) in Section 4.2.1 is the desired one and used as the input of PID controller. The current pose ***F***(*k*) of manipulator end-effector is served as the feedback value of PID controller. The output $V_{PID}(k)$ of PID controller is defined as the end-effector’s velocity to reduce the steady-state error. It is expressed as:(9)$$V_{PID}(k)=K_{p}e(k)+K_{i}\sum_{n=0}^{k}e(k)+K_{d}\left[e(k)-e(k-1)\right]$$

In which ***e***(*k*) = ***X***(*k*) − ***F***(*k*); *K_p_*, *K_i_*, and *K_d_* denote the proportional, intergral, and differential coefficients of PID controller, respectively. The output of PID controller is used to correct the tracking velocity of end-effector in small range, and the large part of tracking velocity is adjusted by the feedforward value obtained by the improved Kalman filter. Therefore, the control velocity of manipulator end-effector can be expressed as:(10)V(k)=VPID(k)+vwg(k)
where $V_{PID}(k)$ and vwg(k) denote the output of PID controller and feedforward value, respectively.

Generally, PID controller will continuously adjust the deviation until it disappears. However, the deviation always exists due to the influence of the manipulator model error and the coordinate transformation error between the camera and global frame. As a result, PID controller will perform frequent adjustments for the fine error when manipulator end-effector approaches the target, which leads to the jitter of manipulator. To solve this problem, the deviation between the desired and feedback pose is pre-processed by the dead zone constraint. If the deviation is too large so that it exceeds the threshold of dead zone constraint, it will stay unchanged to enable the operation of PID controller. When the deviation is small enough, it will be set to zero and the PID controller will stop working. The dead zone constraint and control signal are expressed as Equations (11) and (12), respectively.
(11)$$e(k)=\{\begin{array}{ll} e(k) & \left|e(k)|>|e_{0}\right|\\ 0 & \left|e(k)|\leq |e_{0}\right| \end{array}$$
(12)V(k)={VPID(k)+vwg(k)|e(k)|>|e0|vwg(k)|e(k)|≤|e0|
where ***e***_0_ denotes the threshold of dead zone constraint. It is noted that the setting for the dead zone value is very important. The small value has little effect on reducing manipulator’s jitter. On the contrary, it will cause large control error. In this paper, the dead zone threshold with 0.1 mm is determined according to the control accuracy of manipulator system and the results of pre-experiments.

Specifically, a step response experiment was conducted to analyze the stability of PID controller with dead zone constraint (DPID). The results of the DPID controller and that of the general PID controller for comparison are shown in Figure 8. The desired positions of end-effector in global frame are set at 0.13 s, as marked by the black dotted lines. It can be noted that the adjustment time of DPID controller is smaller than that of the general one. Besides, the motion trajectory obtained by DPID controller is also smoother in the adjustment process, indicating that DPID controller benefits by reducing the jitter of manipulator. This is owed to the DPID controller avoiding frequent adjustment under the dead zone constraint.

#### 4.2.3. Online Tuning of PID Coefficients

In the presence of external disturbance, random noise, and model’s uncertainty, it is hard to track the moving target accurately by the general PID controller [9]. According to the demand of system operation, fuzzy control is used to modify PID controller in this paper, considering its advantages of independent mathematical model, imitating human logical thinking, and strong robustness. According to the general PID controller as expressed in Equation (9), three parameters, *K_p_*, *K_i_*, and *K_d_*, should be tuned by fuzzy tuners. The detailed structure is shown in Figure 5 for the fuzzy control module. There are two inputs for fuzzy controller: the error ***e*** between the pose estimated by the improved Kalman filter and the feedback one of end-effector, and its change rate ***e_c_***. According to fuzzy set rules, the modified values ∆*K_p_*, ∆*K_i_*, and ∆*K_d_* are adjusted on the basis of the initial values *K_po_*, *K_io_*, and *K_do_*. Finally, the fuzzy control module dynamically outputs the changed PID parameters *K_p_*, *K_i_*, and *K_d_* under different operation conditions.

In fuzzy control module, fuzzification is firstly conducted, namely the above five variables are transformed to seven fuzzy linguistic parameters NB (negative big), NM (Negative medium), NS (Negative small), Z (zero), PS (Positive small), PM (Positive medium), PB (Positive big). The corresponding domain of the input and output variables are defined as {−6, −4, −2, 0, 2, 4, 6} and {−3, −2, −1, 0, 1, 2, 3}, respectively. Then, the membership functions (MFs) with triangular distribution are adopted for all variables, and the membership value can be obtained based on the MFs. Considering the demand for accuracy, stability, and overshoot of control system, the fuzzy rules to adjust the PID parameters are determine by the knowledge of the practical experiments, as shown in Table 1. For the large ***e*** and ***e_c_***, larger *K_p_*, *K_i_*, and *K_d_* should be taken to achieve fast tracking. For the medium ***e***, small *K_p_* is used to reduce the overshoot, and *K_i_* should be appropriately selected because it has a significant impact on system response. When ***e*** is small, small *K_p_* and medium *K_i_* are taken to avoid overshoot and reduce steady-state error. For large ***e_c_***, the large value of *K_d_* should be taken to obtain predictive compensation.

According to the above setting and fuzzy rules, the output surface of ∆*K_p_*, ∆*K_i_*, and ∆*K_d_* on the domain are obtained as presented in Figure 9. The modified parameters ∆*K_p_*, ∆*K_i_*, and ∆*K_d_* can be obtained by checking Table 1. Centroid defuzzification method is used to calculate the specific variation of PID parameters from the obtained fuzzy results. Finally, the self-tuning PID parameters are determined by Equation (13).
(13)$$\left\{ \begin{array}{l} \begin{array}{c} K_{p}(k)=K_{po}+\Delta K_{p}(k) \end{array}\\ K_{i}(k)\begin{array}{c} = \end{array}K_{io}+\Delta K_{i}(k)\\ K_{d}(k)=K_{do}+\Delta K_{d}(k) \end{array}\right.$$
where ∆*K_p_*(*k*), ∆*K_i_*(*k*), and ∆*K_d_*(*k*) are the defuzzified variation of proportional, integral, and differential coefficient in the *k*-th control. The initial parameters, including *K_po_* with 28, *K_io_* with 0.008, and *K_do_* with 0.0015, are offline determined by GA-II optimization algorithm, as described in the literature [30].

## 5. Experimental Results and Discussions

In this section, the experiments are performed on the simulation platform to test the dynamic response speed and robustness of the developed controller. Meanwhile, the autonomous operation experiment is conducted to verify the effectiveness of the proposed control system. The experimental scenario is the same as described in Section 3. The pose of manipulator end-effector and moving target are all represented in the global coordinate system, and the tracking error is the difference between the manipulator end-effector and moving target, as expressed in Equation (3).

### 5.1. Dynamic Response Speed Analysis

To test the dynamic response speed of the developed controller, experiments were performed under two conditions of tracking the objects moving at high-speed with 1 m/s and low-speed with 0.6 m/s along *x*-axis, respectively. The results obtained by the developed controller were compared with those obtained by conventional PID, velocity feedforward PID (VFPID), and fuzzy PID (FPID).

Firstly, the tracking error of manipulator end-effector for the low-speed moving target is shown in Figure 10. Taking the results obtained by the developed controller as an example, phase a featured with large changes in tracking error is the transition process of end-effector from the initial point to the near region of the moving target. Phase b denotes that end-effector approaches the near region of the moving target, which is characterized by a small variation in tracking error. Phase c indicates that the stable tracking is achieved, and the capture operation can be performed. From the results in Figure 10a,b, it can be observed that the FPID controller takes less time to reach the stable tracking stage than PID controller due to using the adaptively tuned PID parameters. VFPID controller has a faster response speed than PID and FPID controllers. Compared to the above controllers, the developed controller takes the least time to approach the stable tracking stage. The detailed time spent in phase a and b are presented in Table 2.

Besides, a comparative experiment is also carried out by tracking the high-speed moving object. Figure 11 presents the tracking error under different controllers and the same conclusion can be obtained as the above analysis. Table 3 presents the time spent in phase a and b for the high-speed moving target. It is noted that all the controllers in Table 3 take less time in phase a and b than that in Table 2. That is because the object moves at high speed, which will result in larger tracking error in the tracking process. The output of all the controllers will increase, thus reducing the adjustment time.

### 5.2. Robustness Test under Gauss Noise and Disturbance

To examine the robustness of the developed controller, experiments are performed for the first time changing the target’s position by introducing large Gaussian noise in the measurement equation of Kalman filter, and for the second time introducing an angle disturbance to six joints, as shown in Figure 5. Gaussian noise with zero-mean value and 0.05 covariance is used as the measurement noise from camera. The target’s position along *x* and *y* axis on pipeline in the tracking process using the developed controller is shown in Figure 12.

The tracking error from the initial pose of manipulator end-effector to the stable tracking phase is shown in Figure 13. It can be seen that the proposed VFPIDDF controller has better robustness and a higher response speed than other controllers.

The disturbance with −0.15 rad is introduced to six joint angles between 3.052 s and 3.152 s, as shown in Figure 14. The tracking errors along *x*-axis and *y*-axis are presented in Figure 15a,b, respectively. It is seen that the tracking errors increase sharply under the influence of interference. Take the results obtained by the developed controller as an example, *a_ex_* and *a_ey_* denote the error variation amplitude caused by disturbance, and *t_ex_* and *t_ex_* indicate the adjustment time from appearing disturbance to return to the steady tracking state along *x*-axis and *y*-axis, respectively. Table 4 gives the results of error variation amplitude and adjustment time. It can be noted that through the adjustment of the developed controller, the tracking error not only has smaller amplitude, but also reaches the stable state more quickly than other controllers.

The PID coefficients of the developed controller are adaptively tuned during the tracking process in the presence of disturbance, which are presented in Figure 16. When there are large tracking errors, the velocity feedforward control plays a major role. Thus, a faster response is obtained by the developed controller than the PID controller. At this time, the proportional module plays a minor adjustment role with small proportional coefficient P. Near the stable tracking phase i.e., 1~2 s, the role of velocity feedforward decreases gradually as the tracking of errors decreases. At this time, the coefficient P tends to increase to further reduce the tracking error. During the stable tracking phase, the proportional coefficient P tends to be unchanged for high stability, and the integral coefficient I fluctuates to eliminate the steady-state error. The differential coefficient D is almost zero, because there is no significant position tracking lag under the control of velocity feedforward module.

### 5.3. Autonomous Operation Experiment

To test the practical performance of the proposed control system, the autonomous operation experiment is performed, including tracking and grasping the object moving on pipeline and placing it into the material box. At first, the manipulator receives the output signal of the developed controller to track the moving target. To ensure the real-time performance, the trajectory of manipulator end-effector from the initial point S to a certain grab point G is online, planned based on the proposed strategy, as presented by the dotted line in Figure 17. When the tracking error is within the allowed 5 mm error threshold, the manipulator end-effector performs a grasping operation. Finally, the captured object is placed into the material box along the offline planned trajectory, as shown by the dashed line between point G and E in Figure 17. The trajectory of moving object is the one from start point O to grab point G, as shown by the solid line in Figure 17, and the corresponding tracking error is given in Figure 18.

The key frames of the autonomous operation using 6-DOF manipulator are presented in Figure 19. The manipulator end-effector firstly starts from the starting point and passes through the transition point to reach the tracking point, as shown in Figure 19a–c. When the tracking error is within the allowed 5 mm error threshold, the grasping operation is conducted as shown in Figure 19d,e. The offline placement operation will be performed when the moving target is completely captured. At this time, manipulator end-effector lifts the object, moves it to the pre-placement pose and performs the placement operation, as shown in Figure 19f–h. Finally, manipulator end-effector returns to the initial point represented in Figure 19i and prepares for the next round of autonomous operation. The above experimental results confirm the practicality of the proposed capture and placement control system.

In addition, the autonomous operation experiments with and without using the capture and placement strategy are performed. For the experiments without using the proposed strategy, all the trajectories of manipulator end-effector are online planned. In particular, the trajectory in placement stage is obtained by inverse kinematics, which is different from the one using a proposed strategy. The overall time spent in the experiments with and without using the proposed strategy is recorded, which is denoted by *t_i_* and *t_o_*, respectively. Table 5 presents the result obtained by the developed control system and the traditional ones for comparison, including PID, VFPID, and FPID. It can be noted that the overall time spent in the autonomous operation by the developed control system is less than others. Besides, for all controllers, the time spent in the operation with using the proposed strategy is smaller than that without using the strategy. That is because for the operation without using the proposed strategy, the inverse kinematic calculation is conducted online based on the pre-defined waypoints in the placement process, which will cost more time. The above experiments demonstrate that the developed control system has a high operation efficiency.

## 6. Conclusions

This paper develops a capture and placement control system for 6-DOF manipulator to capture the moving target. An online capture and offline placement strategy is adopted in the developed system, which improves the overall manipulation efficiency compared with the existing control system. In addition, a new PBVS controller is designed by combining the estimated velocity feedforward module and refined PID controller. The motion estimation method combined Kalman filter and interpolation operation is proposed and the accurate and smooth estimation for target’s pose and velocity are thus obtained, which provides velocity value to feedforward module. The PID controller is refined by dead zone constraint to reduce the manipulator’s jitter, and by fuzzy control to adaptively tune PID parameters. Validation experiments prove that the developed controller achieves faster response speed and stronger anti-interference ability compared to the existing PID-based controller. In the future, the autonomous operation for moving target in 3D space will be conducted on the real experimental setup, and the end-to-end visual servo manipulation will be studied to further improve the robustness of the autonomous operation system.

## Figures and Tables

**Figure 1 sensors-22-04836-f001:**
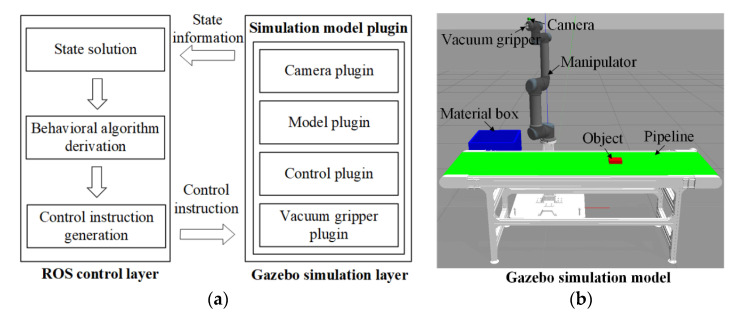
Simulation platform based on ROS–Gazebo. (**a**) Simulation framework. (**b**) Simulation model.

**Figure 2 sensors-22-04836-f002:**
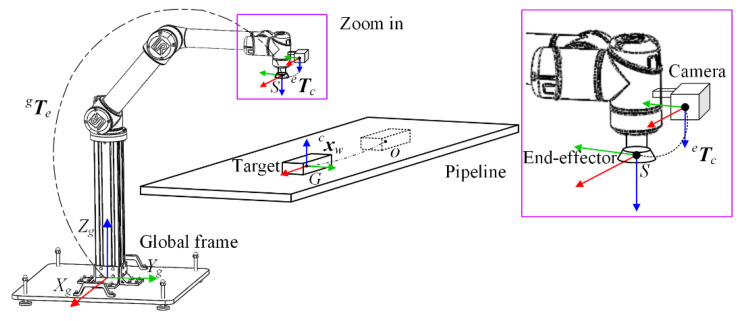
Schematic diagram of capturing target by the visual servoing manipulator.

**Figure 3 sensors-22-04836-f003:**
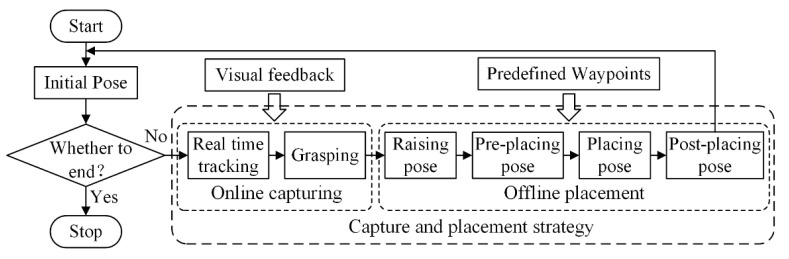
Flowchart of capture and placement strategy for the moving object.

**Figure 4 sensors-22-04836-f004:**
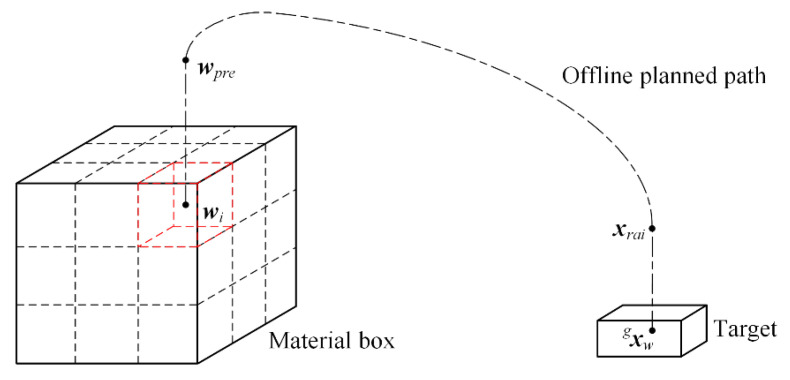
Schematic diagram of offline planning placement path.

**Figure 5 sensors-22-04836-f005:**
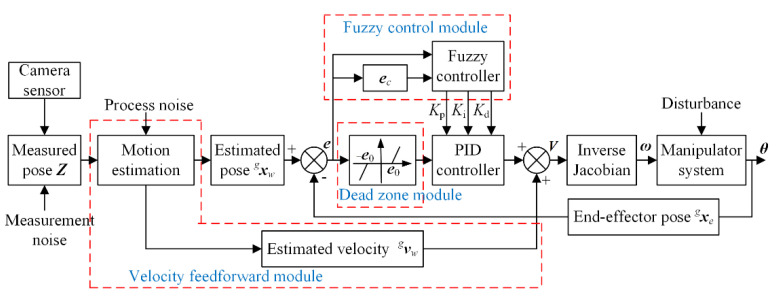
Block diagram of feedforward PID controller with dead zone and fuzzy control module.

**Figure 6 sensors-22-04836-f006:**
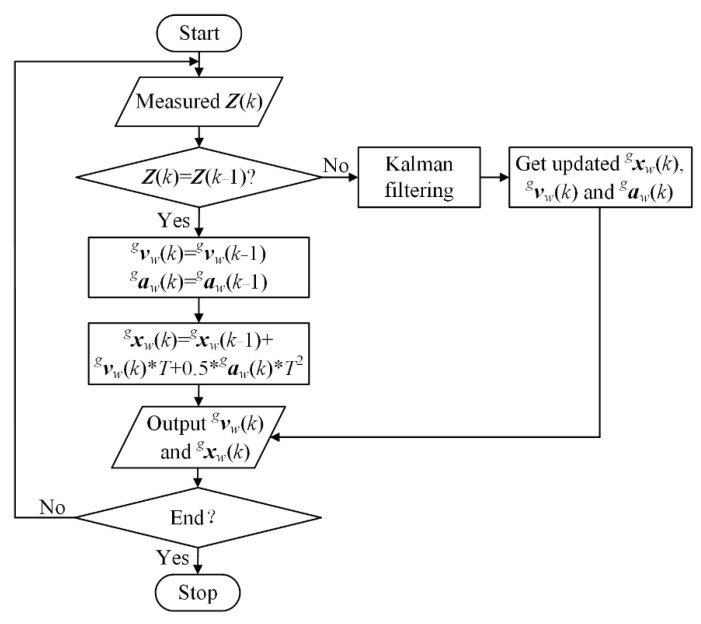
Flow chart of the proposed motion estimation method.

**Figure 7 sensors-22-04836-f007:**
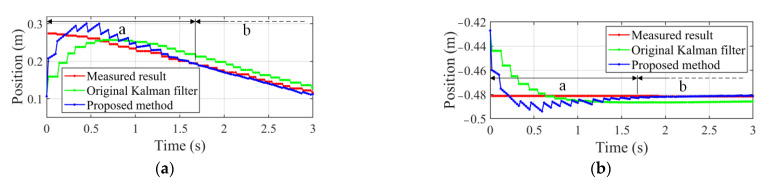
Measured result and estimated ones by original Kalman filter and proposed method. (**a**) Position along *x* axis. (**b**) Position along *y* axis.

**Figure 8 sensors-22-04836-f008:**
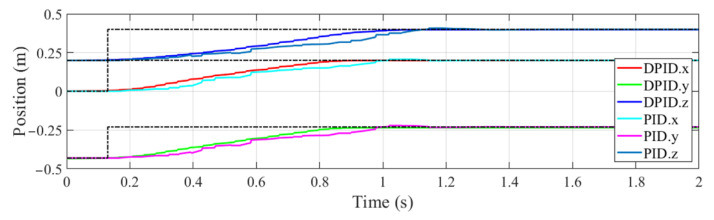
Step response of the position of manipulator end-effector.

**Figure 9 sensors-22-04836-f009:**
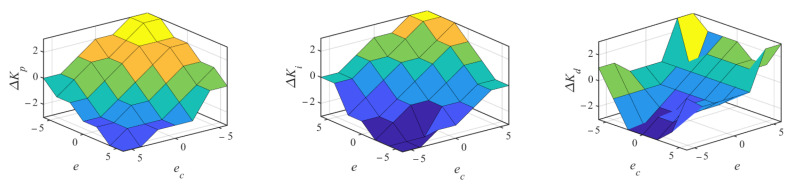
Output surface for self-tuning ∆*K_p_*, ∆*K_i_*, and ∆*K_d_*.

**Figure 10 sensors-22-04836-f010:**
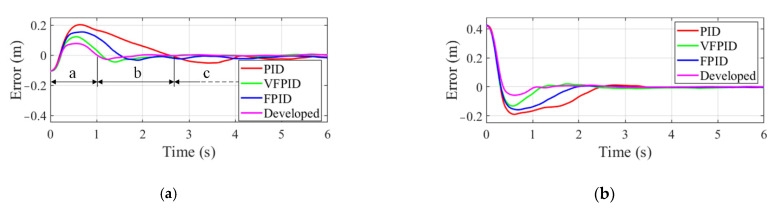
Tracking error of manipulator end effector to low-speed moving target. (**a**) Error along *x* axis. (**b**) Error along *y* axis.

**Figure 11 sensors-22-04836-f011:**
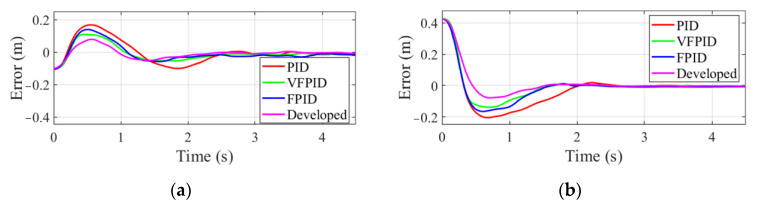
Tracking error of manipulator end effector to high-speed moving object. (**a**) Error along *x* axis. (**b**) Error along *y* axis.

**Figure 12 sensors-22-04836-f012:**
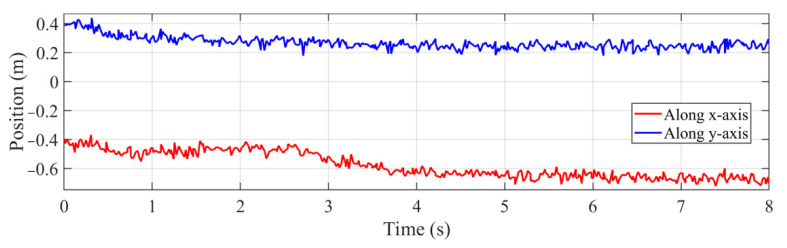
Target’s position obtained by the proposed estimation method with Gaussian noise.

**Figure 13 sensors-22-04836-f013:**
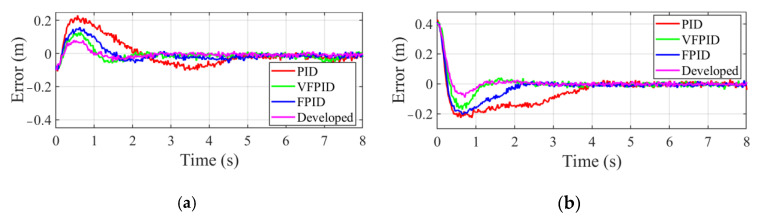
Variation of tracking error in the tracking process with Gaussian noise. (**a**) Error along *x* axis. (**b**) Error along *y* axis.

**Figure 14 sensors-22-04836-f014:**
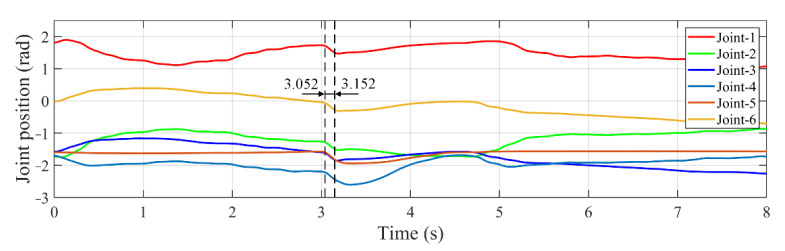
Variation of joint position in the tracking process with −0.15 radian disturbance.

**Figure 15 sensors-22-04836-f015:**
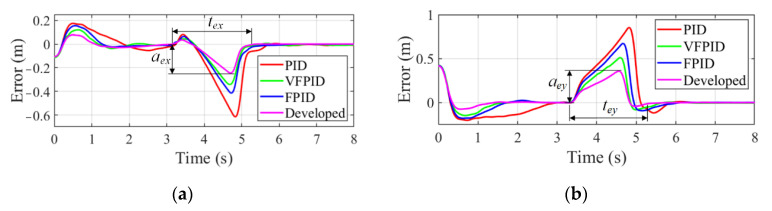
Variation of tracking error in the tracking process with −0.15 radian disturbance. (**a**) Error along *x*-axis. (**b**) Error along *y*-axis.

**Figure 16 sensors-22-04836-f016:**
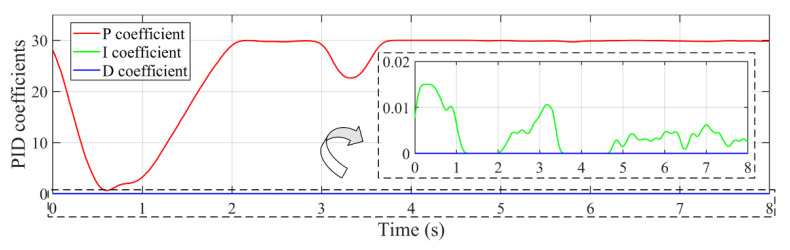
Variation of PID coefficients in the tracking process with −0.15 rad disturbance.

**Figure 17 sensors-22-04836-f017:**
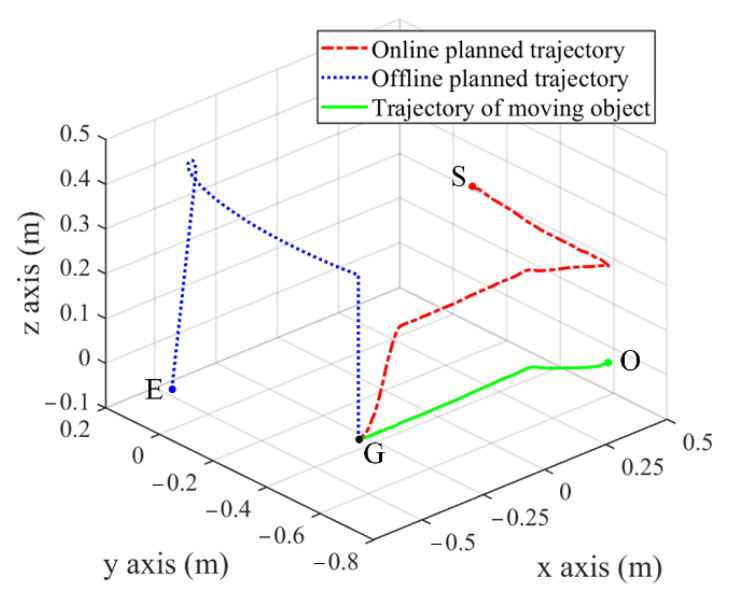
Trajectory of manipulator end-effector and moving object in experiment.

**Figure 18 sensors-22-04836-f018:**
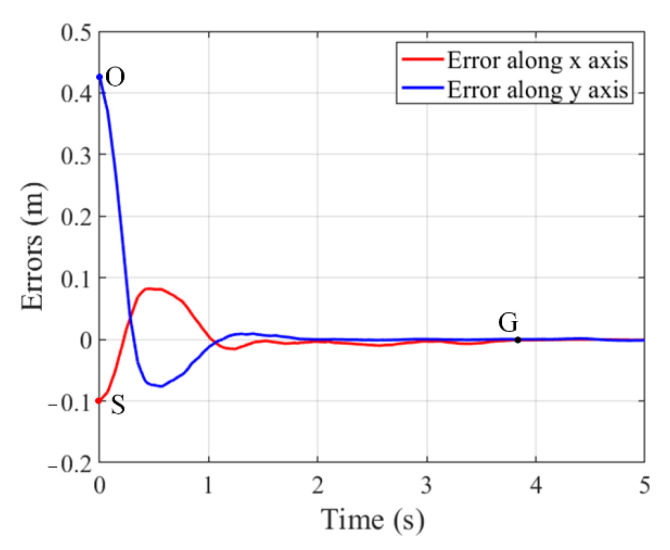
Tracking error of manipulator end-effector to the moving target.

**Figure 19 sensors-22-04836-f019:**
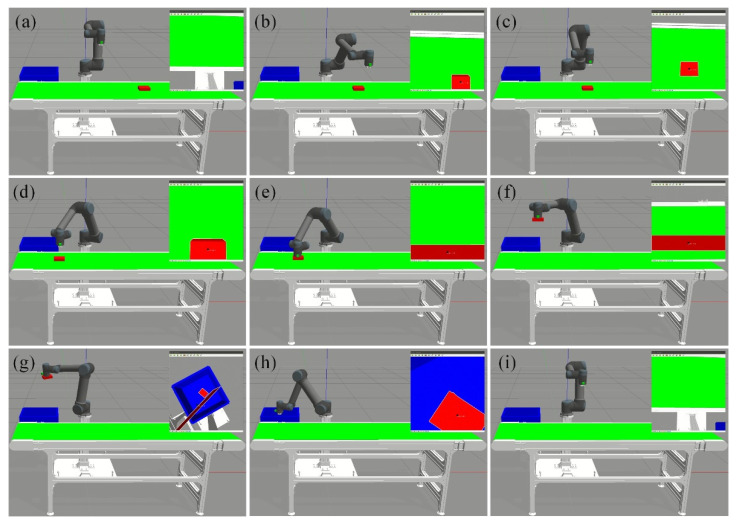
The key frames in the autonomous operation experiment. (**a**) Initial pose. (**b**) Transition phase. (**c**) Start tracking. (**d**) Before grasping. (**e**) Perform grasping. (**f**–**h**) Lift object, perform pre-placement and placement. (**i**) Return to initial pose.

**Table 1 sensors-22-04836-t001:** Fuzzy rules of online tuning (∆*K_p_*, ∆*K_i_*, and ∆*K_d_*).

	*e_c_*	NB	NM	NS	Z	PS	PM	PB
*e*								
NB		(PB, NB, PS)	(PB, NB, NS)	(PM, NM, NB)	(PM, NM, NB)	(PS, NS, NB)	(Z, Z, NM)	(Z, Z, PS)
NM		(PB, NB, PS)	(PB, NB, NS)	(PM, NM, NB)	(PS, NS, NM)	(PS, NS, NM)	(Z, Z, NS)	(NS, Z, Z)
NS		(PM, NB, Z)	(PM, NM, NS)	(PM, NS, NM)	(PS, NS, NM)	(Z, Z, NS)	(NS, PS, NS)	(NS, PS, Z)
Z		(PM, NM, Z)	(PM, NM, NS)	(PS, NS, NS)	(Z, Z, NS)	(NS, PS, NS)	(NM, PM, NS)	(NM, PM, Z)
PS		(PS, NM, Z)	(PS, NS, Z)	(Z, Z, Z)	(NS, PS, Z)	(NS, PS, Z)	(NM, PM, Z)	(NM, PB, Z)
PM		(PS, Z, PB)	(Z, Z, NS)	(NS, PS, PS)	(NM, PS, PS)	(NM, PM, PS)	(NM, PB, PS)	(NB, PB, PB)
PB		(Z, Z, PB)	(Z, Z, PM)	(NM, PS, PM)	(NM, PM, PM)	(NM, PM, PS)	(NB, PB, PS)	(NB, PB, PB)

**Table 2 sensors-22-04836-t002:** The time spent in phase a and b for low-speed moving target.

	PID	VFPID	FPID	Developed
Phase a (s)	2.573	1.098	1.495	1.010
Phase b (s)	5.752	3.121	4.930	2.852

**Table 3 sensors-22-04836-t003:** The time spent in phase a and b for high-speed moving target.

	PID	VFPID	FPID	Developed
Phase a (s)	1.253	1.040	1.101	0.932
Phase b (s)	4.243	2.824	3.982	2.487

**Table 4 sensors-22-04836-t004:** The results of error variation amplitude and adjustment time.

	PID	VFPID	FPID	Developed
*a_ex_* (m)	0.615	0.342	0.415	0.244
*a_ey_* (m)	4.244	2.717	3.98	2.484
*t_ex_* (s)	3.872	2.756	3.392	2.285
*t_ey_* (s)	3.767	2.682	3.317	2.217

**Table 5 sensors-22-04836-t005:** Overall time spent in autonomous operation.

	PID	VFPID	FPID	Developed
*t_i_* (s)	10.541	6.435	8.347	5.825
*t_o_* (s)	11.260	7.129	9.052	6.516

## Data Availability

The data presented in this study are available on request from the corresponding author. The data are not publicly available due to the funded project’s scope of deliverables.

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
