# Peer review of "Autonomous 6-DOF Manipulator Operation for Moving Target by a Capture and Placement Control System"

_sensors, 2022, doi:10.3390/s22134836_

Round 1

Reviewer 1 Report

In my opinion, the manuscript is suitable for publication in Sensors journal but Authors must complete a major revision. Manuscript should be revised according to following comments:

1. Chapter “Introduction” must be improved:
a) the authors present the state of the art in the field of manipulator control and the content of the article (in the last paragraph). However, the authors do not combine these two parts: they do not present how the content of the article will add to the knowledge. Given that this is intended to be a scientific article, authors need to be clear about what is new to the scientific dream in their article.

2. Chapter “Kinematic model of 6-DOF manipulator capturing moving target” must be improved:
a) Figure 1: this drawing is completely illegible. The legibility of details, especially the coordinate systems, should be improved,
b) transformation matrices should be presented in appendix.

3. Chapter “Capture and placement control system” must be improved:
a) Figure 3: Mark the control signal. Noise should not be added to the control signal. Enter the control noise and the measurement noise according to the Kalman filter theory,
b) Figure 5 shows the measured and estimated courses of variables. The authors must present the methodologies of laboratory tests, that is at least:
- diagram of the laboratory stand,
- names and parameters of the measuring apparatus used and used to generate control signals,
c) simulation research methodology should also be presented (software, basic parameters),
d) Kalman filter model must be presented. How were covariance matrices selected?

4. Chapter “Experimental results and discussions” must be corrected:
a) Figures 9, 10, 12 and 14 show the errors. The errors depend on the correctness of the regulators settings. Hence it is necessary to:
- specify the controller settings for which these errors were obtained,
- describe the importance of selecting these settings,
b) subsection 4.3. should be corrected. noise should be added according to the Kalman filter theory (i.e. to the output and to individual state variables)

5. Chapter “Conclusions” must be improved:
a) this chapter must be significantly changed. In its current form, the conclusions are a summary of the work performed. However, there is no clear indication of what, from the scientific point of view, is a new achievement in relation to the existing state of knowledge.

Reviewer 2 Report

The paper proposes a tracking system for a manipulator with a hand-eye behind a moving object. A manipulator movement controller has been developed in which the PID controller parameters are adjusted by a fuzzy logic block. There are a few notes about the work:
1) The authors consider motion control only along the X and Y coordinates. And they do not give results (except for a certain trajectory in Fig. 16) by the movement of the manipulator along the Z axis in time.
2) To reduce computational operations, the authors propose to plan the motion path based on predefined waypoints. However, the implementation of the proposed control is not given.
3) The paper does not provide a procedure for determining the position of an object (blocks “Measured pose Z” and “Estimated pose xw” fig.3) when analyzing an image in a camera.

Reviewer 3 Report

The authors have investigated the "Autonomous 6-DOF manipulator operation for moving target by a capture and placement control system". The study sounds interesting. However, more references may need to be added to enhance the quality and novelty of the work specifically in the introduction section of the manuscript.

For example.

1.Position-based visual servoingin robotic capture of moving target enhanced by Kalman filter

2. Friction Compensation in Robot Manipulator Using Artificial Neural Network

3.Autonomous robotic capture of non-cooperative target using visual servoing and motion predictive control

Secondly, the manuscript should be reviewed and revised by a native English speaker as there are grammatical mistakes in the manuscript. For example, on page 2, lines 89, 90, etc.

Round 2

Reviewer 1 Report

I accept in present form.